# RetCare: Towards Interpretable Clinical Decision Making through LLM-Driven Medical Knowledge Retrieval

Zixiang Wang*
Peking University
Beijing, China
zixiangwang@buaa.edu.cn

Yinghao Zhu*
Peking University
Beijing, China
yhzhu99@gmail.com

Junyi Gao
University of Edinburgh
Health Data Research UK
Edinburgh, Scotland, UK
junyi.gao@ed.ac.uk

Xiaochen Zheng
ETH Zürich
Zürich, Switzerland
xzheng@ethz.ch

Yuhui Zeng
Peking University
Beijing, China
yuhuiz@buaa.edu.cn

Yifan He
Peking University
Beijing, China
heyf@stu.pku.edu.cn

Bowen Jiang
Peking University
Beijing, China
jbw@stu.pku.edu.cn

Wen Tang
Peking University Third
Hospital
Beijing, China

Ewen M. Harrison
University of Edinburgh
Edinburgh, Scotland, UK
ewen.harrison@ed.ac.uk

Chengwei Pan
Beihang University
Beijing, China
pancw@buaa.edu.cn

Liantao Ma†
Peking University
Beijing, China
malt@pku.edu.cn

Ling Wang†
Affiliated Xuzhou
Municipal Hospital of
Xuzhou Medical University
Xuzhou, Jiangsu, China

## ABSTRACT

The integration of Electronic Health Record (EHR) data has greatly advanced clinical decision-making by providing vast amounts of patient information. However, despite significant progress in machine learning models for predicting patient outcomes, these models are rarely used in clinical practice due to their limited interpretability. To address this, we propose RetCare, a workflow that enhances model interpretability by incorporating authoritative medical literature. RetCare leverages the retrieval-augmented generation (RAG) pipeline, utilizing over two million entries from PubMed, combined with the zero-shot reasoning capabilities of large language models (LLM). Our approach focuses on validating machine learning outputs with references from authoritative sources to build clinician trust, developing comprehensive prompting strategies to integrate model outputs with healthcare context, and providing detailed, interpretable reasoning to support clinical decisions. Experimental results on two real-world datasets demonstrate that RetCare significantly improves the accuracy and reliability of model predictions, facilitating more informed and trustworthy clinical decision-making. The code is publicly released at https://github.com/PKU-AICare/RetCare.

## 1 INTRODUCTION

The integration of Electronic Health Record (EHR) data has revolutionized clinical decision making, offering vast repositories of patient information that enhance the quality of care and facilitate informed medical decisions [7, 13]. Despite the significant advancements in machine learning models for predicting patient outcomes and identifying complex patterns within EHR data [5, 25, 29], these models are seldom deployed in clinical practice [24]. One of the primary reasons for this disconnect is the limited interpretability of model outputs [14, 16]. Even models that perform well on

test datasets often provide predictions through logits, which clinicians find difficult to trust [4]. Although some models attempt to offer interpretability by generating personalized feature importance [3, 14, 26], these outputs frequently do not align with the decision making logic of physicians, which is typically based on clinical guidelines and research literature. As a result, physicians may still be unsure of the reasoning behind certain values, leading to skepticism and a lack of trust in these model outputs [13]. Therefore, the challenge lies in **designing an AI workflow that clinicians can reliably work with, providing more reliable sources in the models' outputs that clinicians are willing to accept**.

Through extensive discussions with clinicians, we have learned that they place significant trust in authoritative medical literature. This insight leads us to consider enhancing model interpretability by referencing external authoritative knowledge sources. If model outputs could be supported or even refuted by such sources, clinicians could make more informed and reliable judgments based on the authoritative knowledge. This approach aligns with real-world clinical decision making practices, where doctors consult medical textbooks and relevant literature to support their diagnoses and treatment plans. Motivated by these mirroring processes, we intuitively consider leveraging the Retrieval-Augmented Generation (RAG) pipeline [11] with PubMed [1]'s over two million medical literature entries as a knowledge base, combined with the zero-shot reasoning capacities of Large Language Models (LLMs) as shown in [28], where LLMs have demonstrated their ability to predict patient mortality outcomes with longitudinal EHR data.

While there are previous works that incorporate external auxiliary knowledge from knowledge graphs, clinical notes, etc., to enhance predictive performance, they normally extract and encode this medical knowledge into structured data [6, 9, 12, 20], failing to provide authoritative source knowledge that is relevant to personalized prediction results. Though MedRetrieval [23] attempts

---

*Equal contribution.
†Corresponding author.

to retrieve relevant knowledge from authoritative sources, it only extracts text segments rather than sentence or paragraph-level support, thereby potentially lacking ample semantic information in the patient healthcare context.

To overcome the above-mentioned challenges and limitations, we propose the RetCare workflow, which aims to bridge the gap between AI model outputs and the clinical decision making practices. The implementation of the RetCare workflow embodies our three-fold contributions:

(1) **Authoritative Medical Knowledge with References**: We use external authoritative sources to validate the outputs of machine learning models, including prediction results and feature importance. By providing references from PubMed [1] to support or refute the model's decisions, we emulate the process doctors use to verify information, thereby enhancing the rigor and reliability of clinical decisions. Unlike traditional methods that encode external knowledge into models, our approach offers original sentences and links from PubMed, bolstering clinicians' trust.

(2) **Elaborately Designed Prompts**: We develop comprehensive prompting strategies that incorporate machine learning model outputs, recorded feature statistics, demographics, and disease information into a cohesive patient healthcare context. We use LLMs to extract keywords for medical knowledge retrieval, integrating this retrieved knowledge with the healthcare context to facilitate further decision makings with explanations derived from the LLM's reasoning capabilities.

(3) **Interpretable Reasoning Capacities**: We integrate outputs from multiple EHR models within the patient healthcare context, allowing the LLM to refine a more accurate probability of patient mortality outcomes. We showcase that the LLM-generated explanations serve as interpretability references for clinicians in case studies.

## 2 RELATED WORK

### 2.1 Incorporating External Knowledge for Healthcare Modeling

Numerous studies have explored the integration of clinical background knowledge with EHR data to enhance predictive performance. One primary approach is to leverage medical knowledge graphs (KGs) to enrich the representation learning process of EHR data. For instance, KAME [12] incorporates ontology information throughout prediction process, while MedPath [22] utilizes graph neural networks to capture high-order connections from KGs and integrate them into input representations. MedRetriever [23] combines EHR embeddings with features from target disease documents to retrieve relevant text segments from unstructured medical text, improving health risk prediction and interpretability. KerPrint [21] tackles the issue of knowledge decay across multiple time visits. However, these methods often focus on extracting knowledge into structured representations, potentially overlooking the rich semantic information within the context of knowledge bases, thus limiting the full utilization of incorporated knowledge and highlighting the need for methodologies with semantic reasoning capabilities.

### 2.2 Applying Large Language Models in Healthcare

The rise of Large Language Models (LLMs) as comprehensive knowledge bases [17] has opened up new possibilities for adapting LLMs to EHR data [28]. GraphCare [9] constructs a structured KG from structured EHR data for graph neural network learning but may not fully capture semantic information. Retrieval-Augmented Generation (RAG) technology allows LLMs to expand their knowledge to wider data sources, which is crucial in the healthcare domain [18]. RAM-EHR [20] employs RAG and augments the local EHR predictive model by capturing complementary information from patient visits and summarized knowledge, although it still relies on established graph neural networks for downstream tasks. Interestingly, LLMs have shown promising zero-shot reasoning and prediction capabilities for longitudinal EHR data [28] through direct prompting to generate results, providing a more intuitive approach. However, without authorized external medical knowledge, LLMs may face challenges in updating their internal knowledge, which is particularly important for emerging diseases.

## 3 PROBLEM FORMULATION

Given EHR data ($X$), which includes patient records with various features such as demographics, vitals, lab results, medications, and clinical notes, and an extensive medical knowledge base, such as PubMed articles, the RetCare workflow aims to enhance the interpretability and reliability of model outputs for clinical decision making. The expected outputs of RetCare are two-fold: (1) refined predictions ($\hat{y}'$), (2) explanations with references, which are detailed explanations for the prediction results supported by specific documents from the knowledge base, providing context and justification for the predicted outcomes.

## 4 METHODOLOGY

As depicted in Figure 1, the RetCare workflow processes longitudinal structured EHR data through multiple modules. Initially, EHR models generate logits from EHR data and determine interpretability (feature importance weights). Next, the healthcare context generator formulates a healthcare context based on EHR input and outputs from EHR models, which the keywords generator then uses to produce summarized keywords. These keywords are leveraged by an Information Retriever (MedCPT) to obtain pertinent medical documents from the PubMed knowledge base. Following the retrieval, an LLM integrates the retrieved documents and healthcare context to deliver refined prediction logits and a detailed explanation.

### 4.1 Acquire EHR Models' Outputs

The first step in RetCare workflow is to process longitudinal EHR data to obtain prediction logits and feature importance weights. Given EHR data of a patient $X = [x_1, x_2, \cdots, x_T]^\top \in \mathbb{R}^{T \times F}$, where $T$ is the number of time steps and $F$ is the number of features. Let $\text{Model}_i$ and $\text{Model}_w$ denotes EHR models with and without interpretability inherently, which are used to predict in-hospital mortality risk $\hat{y}$ and learn feature importance weights $\alpha$, as in Eq. 1. For models which do not possess interpretability, we choose the SHAP strategy to output feature importance weights, as in Eq. 2.

$$\hat{y}, \alpha = \text{Model}_i(X) \tag{1}$$

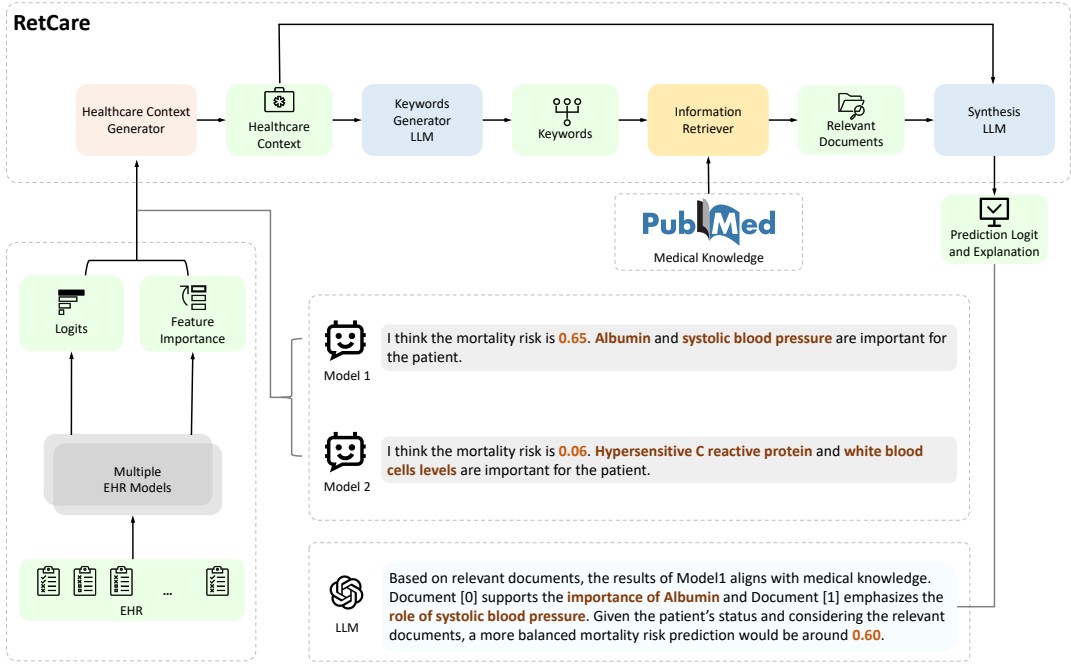

Figure 1: *The overall pipeline of* RetCare.

$$\hat{y} = \text{Model}_w(\boldsymbol{X})$$
$$\boldsymbol{\alpha} = \text{SHAP}(\text{Model}_w, \boldsymbol{X}) \tag{2}$$

## 4.2 Healthcare Context Generation

In RetCare workflow, the LLM analyzes patient information alongside the prediction results from EHR models. To ensure the LLM effectively comprehends the key information, it is essential to use a healthcare context generator integrating critical data into a coherent context, which is accomplished through a rule-based algorithm. The healthcare context is segmented into four distinct parts:

- Basic Information: details including the patient's gender, age, original disease, etc.
- Longitudinal Structured EHR Data: multiple strings of feature values separated by commas, each string corresponding to a feature.
- Data-driven interpretability evidence: important features that EHR models focus on and related data, such as the mean values for surviving and deceased patients, the standard ranges of these features.
- Information of similar patients: the basic information and outcomes (i.e., recover or decease) of similar patients.

The example of generated healthcare context is in the case studies section's Figure 3(a).

## 4.3 Keywords Generation

The complete healthcare context contains a amount of redundant information, which is useful for LLM understanding the patient's status but may reduce the retrieval recall rate. To address this, we extract the basic information and data-driven interpretability evidence parts to form a sub healthcare context, as these parts contain rich semantic information crucial to diagnosis. The keywords generator then extracts keywords from this sub healthcare context to create a query that retrieves relevant medical knowledge more effectively. This is achieved by instructing the LLM via a prompt to extract the most critical and task-relevant information. The prompt is illustrated in Figure 2(a).

## 4.4 Information Retrieval

We retrieve authoritative medical knowledge from the PubMed database and employ a BERT-based sentence transformer model to retrieve and rerank authoritative literature results. Specifically, we calculate the embeddings of the keywords and match them with the embeddings of the medical knowledge via cosine similarity. We select the top $K$ literature entries, which further serve as a part of the healthcare context with authorized references.

## 4.5 Large Language Model Reasoning

In the final step, the retrieved authoritative literature and the healthcare context are provided as complete context input to an LLM. We employ an output prompt template to format the responses of the LLM. The LLM is instructed to rely solely on the retrieved literature for its analysis, ensuring that any conclusions drawn are supported by the literature. Moreover, the LLM is instructed to refine the outputs of the multiple EHR models to $\hat{y}'$ and provide explanations. The prompt is shown in Figure 2(b).

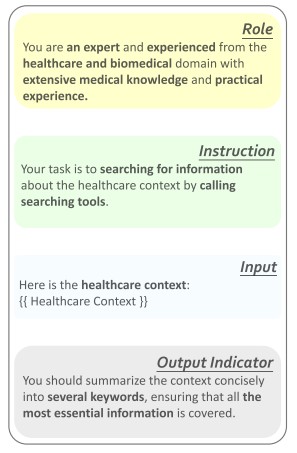

(a) Prompt for keywords generation  (b) Prompt for LLM's final reasoning

**Figure 2:** *Abbreviated prompt templates used in* `RetCare`.

## 5 EXPERIMENTAL SETUPS

### 5.1 Experimented Datasets and Adopted Medicial Knowledge Base

We adopt two real-world datasets: ESRD [13] and CDSL [8] dataset.

The ESRD dataset consists of 656 Peritoneal Dialysis patients with 13,091 visit records, spanning over 12 years, from January 1, 2006, to January 1, 2018, including patients' baseline data, longitudinal visit records, and outcomes.

The CDSL dataset originates from the HM Hospitales EHR system and contains anonymized records of 4,479 patients admitted with a diagnosis of COVID-19 or suspected COVID-19 infection. The dataset includes heterogeneous medical features such as detailed information on diagnoses, treatments, admissions, ICU admissions, diagnostic imaging tests, laboratory results, and patient discharge or death status.

We follow the benchmark preprocessing pipeline as established in the studies [7, 27]. The statistics of dataset split and label distribution for the two datasets are shown in Table 1.

**Table 1:** *Statistics of ESRD and CDSL datasets after preprocessing.* **The number and proportion for labels are the percentage of the label with value** 1. *Out.* **denotes Mortality Outcome.**

| Dataset | Split | Samples | Label$_{Out.}$ |
|---------|-------|---------|----------------|
| ESRD | Train | 458 (70.00%) | 134 (29.26%) |
| | Val | 66 (10.00%) | 19 (28.79%) |
| | Test | 20 (3.05%) | 10 (50%) |
| CDSL | Train | 2978 (70.00%) | 378 (12.69%) |
| | Val | 426 (10.00%) | 54 (12.68%) |
| | Test | 20 (0.47%) | 10 (50%) |

We adopt PubMed [1], a comprehensive database containing over two million medical literature titles, abstracts, and links, as an authoritative knowledge base to be incorporated. It is widely leveraged in medical large language model's pretraining phase and RAG services [2].

## 5.2 Utilized Models

We select three widely used machine learning or deep learning models that perform well on the two datasets and offer interpretability: Logistic Regression and ConCare [14].

We employ GPT-4o [15], one of the most powerful large language model, as our primary reasoning model to extract keywords and generate the final responses.

MedCPT [10] is a Contrastively Pre-trained Transformer for zero-shot biomedical information retrieval (IR), trained on 255 million PubMed click logs, achieving state-of-the-art performance in multiple biomedicalt IR asks.

## 5.3 Implementation Details

For machine learning or deep learning model baselines illustrated in performance table, we train these models on a server equipped with Nvidia RTX 3090 GPU and 128GB RAM. The software environment is CUDA 12.2, Python 3.11, PyTorch 2.0.1, PyTorch Lightning 2.0.5. We use AdamW optimizer. All models are trained via 50 epochs over patient samples on the training set, and the early stop strategy monitored by AUPRC with 10 epochs is applied. The $K$ for reranking is set to 16. The code for information retrieval in our work is based on MedRAG [19] at https://github.com/Teddy-XiongGZ/MedRAG.

## 6 EXPERIMENTAL RESULTS AND ANALYSIS

### 6.1 Experimental Results

Table 2 presents the benchmarking performance of `RetCare`. Within our tailored workflow, `RetCare` consistently outperforms LR and ConCare in terms of accuracy and F1 across both datasets. In the ESRD dataset, `RetCare` excels in all evaluation metrics, with relative improvements in accuracy and F1 reaching 20.00% and 17.11% respectively. In the CDSL dataset, `RetCare` maintains a competitive edge in accuracy and F1 score, although LR achieves slightly better results in AUPRC and AUROC.

**Table 2:** *Benchmarking performance of* `RetCare` *in ESRD and CDSL datasets on in-hospital mortality prediction task.* **ACC. denotes Accuracy.**

| Methods | ESRD | | | |
|---------|------|------|------|------|
| | ACC. (↑) | AUPRC (↑) | AUROC(↑) | F1 (↑) |
| LR | 0.75 | 0.90 | 0.87 | 0.67 |
| ConCare | 0.75 | 0.87 | 0.85 | 0.76 |
| RetCare (LR + Concare) | **0.90** | **0.94** | **0.92** | **0.89** |

| Methods | CDSL | | | |
|---------|------|------|------|------|
| | ACC. (↑) | AUPRC (↑) | AUROC(↑) | F1 (↑) |
| LR | 0.60 | **0.89** | 0.87 | 0.43 |
| ConCare | 0.70 | **0.89** | **0.90** | 0.67 |
| RetCare (LR + Concare) | **0.80** | 0.88 | 0.89 | **0.80** |

### 6.2 Case Studies

To intuitively show how the `RetCare` supports or refutes interpretability of EHR models based on external authoritative medical knowledge, we present examples in Figure 3 and Figure 4(b) .

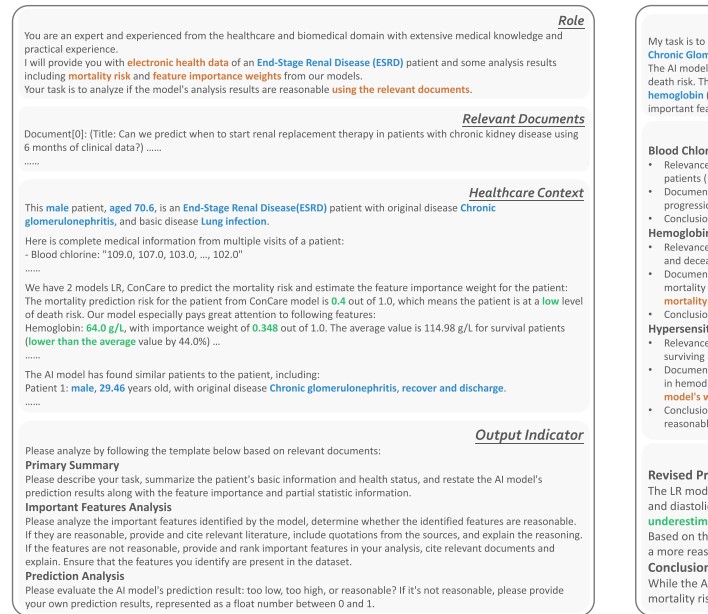

(a) Input of the case on ESRD dataset          (b) Output of the case ESRD dataset

**Figure 3:** *A case study of* `RetCare` *on ESRD dataset.*

As shown in Figure 3(a), the LLM is required to play a role of expert in healthcare and to analyze EHR data of an ESRD patient, utilizing relevant documents that offer theoretical support and background knowledge. The patient's medical background, including specific health conditions and AI model predictions, is detailed to form the basis of the analysis. In Figure 3(b), GPT-4o initially provides a primary summary of the patient's condition and the AI models' mortality risk predictions, highlighting the key features identified by the models. The LLM then evaluates the significance of these features, discussing their relevance and support from the provided documents. The analysis reveals that blood chlorine, emphasized by LR, is not supported by the documents, whereas hemoglobin, a key factor identified by ConCare, is well-supported. Additionally, the low level of hypersensitive C-reactive protein noted by ConCare may conflict with its attention. Finally, the LLM assesses the AI models' predictions and refine a more accurate logit, which is consistent with the patient's outcome (desease), although two models both predict a low mortality risk.

Figure 4 shows another case study on the CDSL dataset. In this case, LR and ConCare provide very different prediction results. After the primary summary, the LLM reveals that the ConCare model's emphasis on oxygen saturation is reasonable and well-supported by multiple studies, whereas other critical features such as elevated LDH levels and age are also significant. The LR model's mortality risk prediction is deemed too low while the ConCare model's prediction aligns more closely with the patient's condition. Finally, the LLM concludes that, given the patient's advanced age and significantly low oxygen saturation, a mortality risk prediction of 0.80 is more accurate and reasonable.

# 7 CONCLUSIONS

`RetCare` presents a comprehensive workflow that addresses the critical need for interpretability and reliability in AI-assisted clinical decision-making. By integrating the Retrieval-Augmented Generation (RAG) pipeline with extensive medical literature and leveraging the reasoning capabilities of large language models (LLMs), `RetCare` provides clinicians with trustworthy and interpretable predictions. The approach validates machine learning results with credible references, employs robust prompting techniques to contextualize model outputs within the healthcare setting, and delivers clear, interpretable explanations that make the AI's reasoning transparent to clinicians. Experimental results on two real-world datasets demonstrate `RetCare`'s significant improvements in accuracy and reliability. `RetCare` paves the way for greater acceptance and integration of AI tools in clinical settings, ultimately leading to enhanced patient care and outcomes.

# 8 DISCUSSION ON ALIGNING RETCARE WITH AIDSH WORKSHOP SCOPE

The "Artificial Intelligence and Data Science for Healthcare (AIDSH)" workshop is a pioneering platform that explores the amalgamation of AI and data science within the healthcare sector. It is particularly focused on the confluence of data-centric AI and people-centric healthcare (PCHC), striving to surmount the challenges that hinder the full realization of AI's potential in healthcare.

Our paper, `RetCare`, harmonizes with the AIDSH workshop's research scope in several fundamental respects:

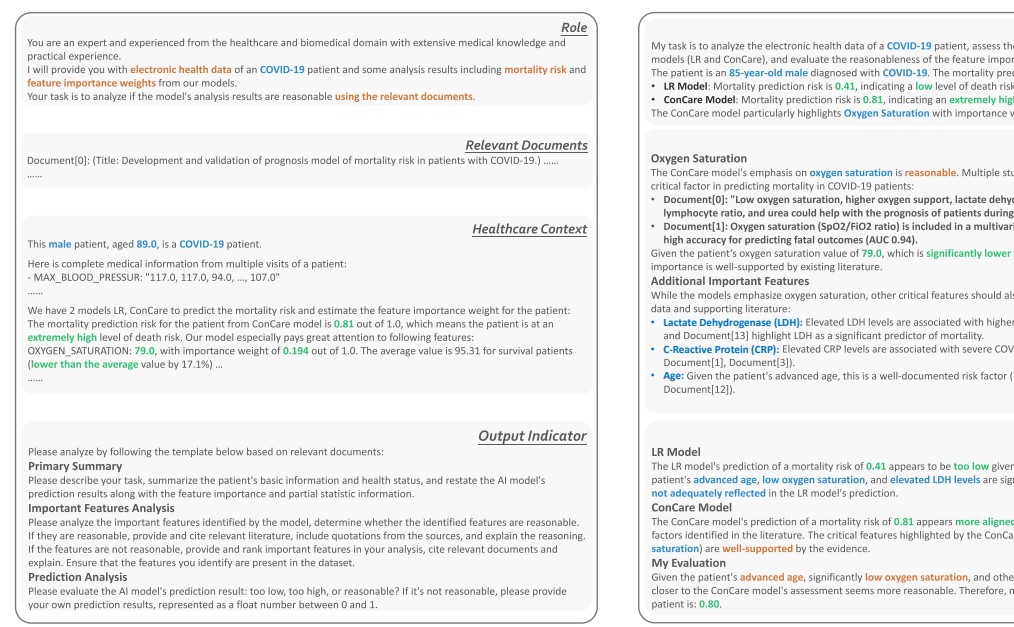

(a) Input of the case on CDSL dataset  (b) Output of the case on CDSL dataset

**Figure 4:** *A case study of* `RetCare` *on CDSL dataset.*

- **Enhancing Model Interpretability and Building Trust in AI-Driven Decisions**: Our approach aligns with the workshop's objective of facilitating Data-Centric AI with insights from People-Centric Healthcare. `RetCare` is designed to address the critical issue of model interpretability in AI-driven clinical decision-making. By integrating authoritative medical literature through the Retrieval-Augmented Generation (RAG) pipeline and leveraging Large Language Models (LLMs) for zero-shot reasoning, `RetCare` provides a workflow that validates AI outputs with references which is a key focus of the workshop's "Explainable AI Models for Trustworthy Healthcare Decisions" track.

- **Empowering Clinical Decision-Making and Facilitating Communication**: The `RetCare` workflow, with its emphasis on detailed and interpretable reasoning, supports clinical decisions in a manner that is consistent with real-world practices. It facilitates effective communication between AI models and healthcare providers. This user-centric approach is in line with the workshop's focus on applications that actively engage individuals in managing health data.

- **Promoting Education**: By publicly releasing the code, `RetCare` contributes to the workshop's mission of promoting transparency in AI applications for healthcare. This openness supports educational initiatives and the development of trustworthy AI tools.

In summary, `RetCare` not only advances the field of interpretable clinical decision-making but also fits seamlessly within the research scope of the AIDSH workshop. It embodies the workshop's vision of creating AI tools that are both data-centric and people-centric, offering a contribution towards building trustworthy and transparent AI applications in healthcare.

## ACKNOWLEDGMENTS

This work was supported by the National Natural Science Foundation of China (U23A20468), and Xuzhou Scientific Technological Projects (KC23143). Junyi Gao acknowledges the receipt of studentship awards from the Health Data Research UK-The Alan Turing Institute Wellcome PhD Programme in Health Data Science (grant 218529/Z/19/Z).

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
