# OpenReview forum: "RetCare: Towards Interpretable Clinical Decision Making through LLM-Driven Medical Knowledge Retrieval"
_KDD.org/2024/Workshop/AIDSH — KDD-AIDSH 2024 Poster_

### Official Review · Reviewer_F4CR · 2024-06-16
**Nice study with RAG for Interpretable Clinical Decision Making**

**Rating:** 9
**Confidence:** 3

**Review:**

Summary: The paper proposes a novel approach to enhance the interpretability of clinical decision-making systems. The RetCare workflow integrates authoritative medical literature using a Retrieval-Augmented Generation (RAG) pipeline and zero-shot reasoning capabilities of large language models (LLMs). By linking model outputs with validated medical knowledge, RetCare aims to build clinician trust and deliver interpretable, reliable predictions.

Strengths:RetCare's use of the RAG pipeline to incorporate authoritative medical literature directly into the decision-making process is innovative. This approach leverages vast databases like PubMed to validate AI predictions, enhancing trust and transparency.

Besides, paper presents comprehensive experiments using real-world datasets to validate the effectiveness of RetCare. The improvements in accuracy and reliability, as shown through comparative analysis with existing models, are statistically significant and well-documented.

Weaknesses: I am worried about that the effectiveness of RetCare is heavily dependent on the quality and relevance of the medical literature it retrieves. Inaccuracies in these external sources could mislead the system, leading to incorrect predictions. Maybe RetCare needs to evaluate and filter the literature first, then using the correct ones to do the RAG.

Besides, the paper does not address how the system scales with increasingly large datasets or in environments with diverse healthcare practices and varying access to medical literature.

---

### Official Review · Reviewer_ejcy · 2024-06-20

**Rating:** 7
**Confidence:** 4

**Review:**

### Brief Summary

The paper introduces RetCare, a workflow designed to enhance the interpretability and reliability of clinical decision-making through the integration of Electronic Health Record (EHR) data with authoritative medical literature. RetCare utilizes the Retrieval-Augmented Generation (RAG) pipeline, harnessing over two million entries from PubMed and the zero-shot reasoning capabilities of Large Language Models (LLMs). The approach aims to build clinician trust by validating machine learning outputs with references from authoritative sources. It develops comprehensive prompting strategies to integrate model outputs with healthcare context and provides detailed, interpretable reasoning to support clinical decisions. The paper demonstrates RetCare's significant improvements in accuracy and reliability through experimental results on two real-world datasets.

### Strengths
- A Timely and Innovative Integration: RetCare's integration of RAG with PubMed and LLMs is an innovative approach to enhancing model interpretability in clinical decision-making.
- Comprehensive Prompting Strategies: The development of comprehensive prompting strategies that contextualize model outputs within the healthcare setting is a significant contribution.
- Empirical Validation: The paper provides empirical validation on two real-world datasets, showing improvements in accuracy and reliability, which is a strength of the research.

### Weaknesses
- The code is not available yet.
- The experimental verifications are still limited. I would like to suggest considering larger datasets and diverse tasks.

---

### Decision · Program_Chairs · 2024-06-28

Accept (Poster)